

# Drought disrupts atmospheric carbon uptake in a Mediterranean saline lake

Ihab Alfadhel[1,2], Ignacio Peralta-Maraver[1,2,3]\*, Isabel Reche[1,3], Enrique P. Sánchez-Cañete[2,4,5], Sergio Aranda-Barranco[1,4], Eva Rodríguez-Velasco[1,3], Andrew S. Kowalski[2,4,5], Penélope Serrano-Ortiz[1,4]\*.

*Authors adress:*

[1]Departamento de Ecología, Facultad de Ciencias, Universidad de Granada, Granada, Spain.

[2]Instituto del Agua, Granada, Universidad de Granada, Spain.

[3]Research Unit Modeling Nature (MNat), Universidad de Granada, Granada, Spain.

[4]Instituto Interuniversitario de Investigación del Sistema Tierra en Andalucía (IISTA), Universidad de Granada, Spain.

[5]Departamento de Física Aplicada, Universidad de Granada, Granada, Spain.

*Correspondence to*: Ignacio Peralta-Maraver (*peraltamaraver@ugr.es*) and Penélope Serrano-Ortiz (*penelope@go.ugr.es*).

**Abstract:** Saline inland lakes play a key role in the global carbon cycle, acting as dynamic zones for atmospheric carbon exchange and storage. Given the global decline of saline lakes and the expected increase of periods of drought in a climate change scenario, changes in their potential capacity to uptake or emit atmospheric carbon are expected. Here, we conducted continuous measurements of $CO_2$ and $CH_4$ fluxes at the ecosystem scale in a saline endorheic lake of the Mediterranean region over nearly 2 years. Our focus was on determining net $CO_2$ and $CH_4$ exchanges with the atmosphere under both dry and flooded conditions, using the eddy covariance (EC) method. We coupled greenhouse gas flux measurements with water storage and analyzed meteorological variables like air temperature and radiation, known to influence carbon fluxes in lakes. This extensive data integration enabled the projection of the net carbon flux over time, accounting for both dry and wet periods on an interannual scale. We found that the system acts as a significant carbon sink by atmospheric $CO_2$ uptake in wet conditions, with uptake ceasing in periods of drought. Moreover, increased air temperatures during wet phases slightly decrease the



$CO_2$ uptake efficiency. Regarding $CH_4$, we measured uptake rates that exceeded those of
well-aerated soils such as forest soils or grasslands. Additionally, we observed that $CH_4$
uptake during dry periods was nearly double that of wet periods. However, the absence
of continuous data prevented us from correlating $CH_4$ uptake processes with potential
environmental predictors. Our study challenges the widespread notion that wetlands are
universally greenhouse gas emitters, highlighting the significant role that endorheic saline
lakes can play as natural sink of atmospheric carbon. However, our work also underscores
the vulnerability of these ecosystem services in the current climate change scenario,
where drought episodes are expected to become more frequent and intense in the coming
years.

**Keywords:** Intermittent saline lake, eddy covariance, greenhouse gas fluxes, ecosystem
metabolism, Mediterranean shallow lake

## 1. Introduction

Saline inland lakes are diverse and play a crucial role in the global carbon cycle, serving
as dynamic zones for carbon dioxide exchange with the atmosphere (Li et al. 2022; Liao
et al. 2024) and long-term sinks of organic and inorganic carbon (Anderson and Stedmon
2007; Song et al. 2013; Li et al. 2017). In limnology, however, the ecological importance
of these systems has only recently been recognized, despite they account for
approximately 44% of global lake volume and 23% of lake surface area (Messager et al.,
2016). Saline lakes vary in size from ephemeral ponds to extensive deep-water bodies,
such as the Caspian Sea (Eugster and Hardie, 1978). These lakes are characterized by
salinity levels that exceed 3 parts per thousand and are notably isolated from a direct
marine influence (Williams, 2002; Wang et al., 2018). They are found in endorheic
(hydrologically landlocked) basins across a wide array of climates, spanning cold to
warm/hot arid regions in all continents, including Antarctica (Williams, 2002; Wang et
al., 2018). As terminal points in many hydrological networks, they collect not only
significant amounts of salts but also nutrients and organic and inorganic carbon
(Anderson and Stedmon 2007; Song et al. 2013; Batanero et al. 2017; Li et al. 2017; Liao
et al. 2024).

The first estimations about the role of saline lakes on global carbon fluxes

suggested that these lakes might function as hotspots for the $CO_2$ emission (Duarte et al.,



2008). However, more recent works point out that saline lakes have lower partial pressure
of $CO_2$ (pCO2) than freshwater lakes (Wan et al. 2017) and some systems appear to
uptake $CO_2$ during the winter (Li et al., 2022) or annually (Yang et al. 2021). Therefore,
more seasonal studies on $CO_2$ fluxes in saline lakes are need it to understand the
conditions when these systems behave as sink or sources of $CO_2$. Variations in $CO_2$ and
$CH_4$ flux estimates across different studies of water bodies are primarily due to the highly
variable data obtained from discrete sampling (Li et al. 2022) or because of differences
in sampling seasons at the intra-annual scale (Liao et al. 2024). Meanwhile, gathering
continuous time series data on $CO_2$ and $CH_4$ sequestration and emission fluxes over years
is needed for accurate assessment of the net carbon balance in inland water systems
(Martínez-García et al. 2024). Nevertheless, long-term, uninterrupted, and direct
monitoring of greenhouse gas flux dynamics at the ecosystem level is relatively scarce in
aquatic ecosystems, and this is particularly true for saline lakes. To the best of our
knowledge, only a couple of studies have reported continuous year-round direct
measurements at the ecosystem scale for $CO_2$ fluxes (Yang et al. 2021; Li et al. 2022).
However, saline lakes' characteristics differ with latitude (Hammer 1986) and could have
very different behaviors regarding carbon exchanges depending on climate conditions.

The carbon hydrochemistry in permanent saline lakes, especially in mountainous

and Arctic latitudes such as the Tibetan Plateau or Svalbard is largely influenced by
surface ice formation (Anderson et al., 2004; Rysgaard et al., 2012, 2013; Wu et al., 2014;
Yan and Zheng, 2015). In contrast, saline lakes in arid and semi-arid endorheic basins,
including Mediterranean climates, are typically shallow, often ephemeral, and/or
hypersaline due to evaporation exceeding precipitation (García et al., 1997; Batanero et
al. 2017; Saccò et al., 2021). The lower depths and higher surface-to-volume ratio, driven
by drought conditions, induce significant physicochemical fluctuations in these saline
inland water bodies, spanning from diurnal to interannual scales (Comin et al. 1990;
García and Niell 1991; García et al. 1997; Batanero et al. 2017). Consequently, the
precipitation regime and subsequent changes in groundwater levels determine the ecology
of saline lakes in arid and semi-arid regions. However, research on the interannual
variability of carbon fluxes in saline lakes affected by seasonal flooding and drought is
lacking. This knowledge gap urgently requires focused research to elucidate the impacts
of climatic variability on the carbon dynamics of these ecosystems, which have been
identified as particularly vulnerable to climate fluctuations (Tweed et al., 2011).
Furthermore, recent studies highlight a global decline in lake water storage in most





endorheic basins and in the Sahara, Arabia, Southern Europe basin in particular (Wang et
al. 2018), a situation expected to worsen with more severe droughts in a climate change
scenario, leading to lower water levels and prolonged desiccation periods (Wurtsbaugh et
al., 2017; Hassani et al., 2020).
In this study, we carried out continuous and interannual measurements of $CO_2$ and
$CH_4$ fluxes at the ecosystem level in the saline lake Fuente de Piedra using the Eddy
Covariance (EC) method. Serving as a model of a Mediterranean shallow saline lake, it
is characterized by sporadic episodes of water retention but predominantly dry during the
summer. The objectives of our work are a) to quantify carbon exchanges during the dry
and the flooded conditions, determining its role as a carbon source or sink, b) to evaluate
main drivers promoting carbon exchange behaviors, and finally c) to model the annual
net carbon flux of the system as a function of its meteorological drivers. Our research
aims to enhance our understanding of the carbon dynamics and the impacts of climate
change on the net carbon balance in Mediterranean intermittent endorheic lakes.

## 2. Material and methods


### Study Site


Fuente de Piedra is a shallow and saline lake located in an endorheic basin in the province
of Málaga, Andalusia, Spain (37.11 N, -4.77 W, elevation 410 m: **Fig 1**). It spans
approximately 17 km$^2$, measuring 6.8 km in length and 2.5 km in width, with a maximum
depth of 1.5 meters. We take advantage of Fuente de Piedra Lake's inclusion in the
Ramsar Convention in 1983. This designation ensures a rich history monitoring of water
storage and the meteorological drivers discussed in this article. Such a comprehensive
dataset allows for the back projection of the net carbon flux of the system over time,
incorporating both dry and wet periods at an interannual scale. This lake is recognized as
a vital habitat within a protected wetland at various levels—regional (as a natural reserve),
European (designated as a special bird protection area), and international (acknowledged
as a Ramsar site)—and offers an exemplary nesting ground for the pink flamingo
(*Phoenicopterus roseus*), largely due to its shallow waters. Among primary producers,
diatoms constitute the largest fraction of primary producers of the phytoplanktpn all
through the year, being dominated by *Hantzschia amphioxys*, *Amphora coffeaiformis*,
*Stauronensis amphioxys*, *Cocconeis placentula*, *Entomoneis* sp. and several species of
*Navicula* and *Nitzchia* sp. (García and Niell, 1993).



Salinity levels in the lake vary significantly, ranging from oligosaline (5 ppt) to
hypersaline conditions (> 200 ppt), influenced by the annual hydrological cycle (Batanero
et al. 2017). This cycle is delineated into two distinct phases: a pooling phase during
autumn and winter (December to March), and an evaporative and drought phase spanning
spring and summer (April to November). The lake primarily receives water from
groundwater inflow (Rodríguez- Rodríguez et al. 2006), complemented by contributions
from two streams (**Fig. 1**) and surface runoff from surrounding farmlands. Notably, the
stream entering from the northeast adds nutrients. However, sediment samples distributed
across the lake and analyzed through combustion (Heiri et al., 2001) showed it to be
homogeneous in organic carbon (0.21 ±0.069 mg C), nitrogen (0.015±0.004 mg N) and
the C:N ratio (14.4± 2.26).

***Field measurements of greenhouse gas fluxes and meteorological drivers.***
We employed the eddy covariance method to quantify the exchanges of $CO_2$, $CH_4$, and
energy (sensible and latent heat) every 30 min from August 2021 to May 2023. Thus,
eddy covariance system was operated for more than 21 months, including two dry periods
in summer. An open-path eddy covariance (EC) system was strategically positioned atop
a tower, 3.1 meters above ground level, on the western bank of the lake (**Fig.1**). This setup
included two open-path infrared gas analyzers: the LI7500 for $CO_2$ and water vapor, and
the LI7700 for $CH_4$ (LICOR Inc., Lincoln, NE, USA). Wind vector components (u, v, w)
and sonic temperature were accurately measured using a sonic anemometer (R.M. Young
81000V, Traverse City, MI, USA). Both instruments recorded the data at a frequency of
10Hz.
In addition to gas measurements, we measured a comprehensive suite of
environmental and soil state variables every 10 seconds to capture the conditions over a
representative ground surface area and collected every 30 min average by a data logger
(CR1000, Campbell Scientific, Logan, UT, USA). A quantum sensor (LI190, Lincoln,
NE, USA) was utilized to measure the vertical component of the incoming photosynthetic
photon flux density (PPFD) at a height of 2.9 meters. Air temperature ($T_a$) and relative
humidity (RH) were monitored using a thermohygrometer (HMP 45C, Campbell
Scientific, Logan, UT, USA). Net radiation ($R_n$) was quantified by a net radiometer (NR
Lite, Kipp and Zonen, Delft, Netherlands). Soil heat flow (G) calculations were facilitated
by one heat flux plate (HFP01SC, Hukseflux, Delft, Netherlands) placed at 8 cm depth,
complemented by three pairs of soil temperature probes (TCAV, Campbell Scientific,



Logan, UT, USA) situated at depths of 4 cm and lateral distances of 3.20 m, 6.34 m, and
8.90 m from the tower.

The groundwater level (GWL) was monitored daily using a piezometer situated

within a well in the salt flats, approximately 2 kilometres south of the EC tower and on
the opposite side of the lake (37.1071° N, -4.7631° W). Furthermore, data on daily
precipitation (PPT), air temperature, and incident solar radiation (spanning wavelengths
from 350 to 1100 nm) were acquired from a meteorological station located adjacent to
Fuente de Piedra Lake, in Sierra Yeguas (37.1383° N, -4.8358°; 467 m.a.s.l.). The tower
setup and instruments were maintained (mainly cleaning lenses of the open path sensors)
every two weeks.

***Greenhouse gas flux data processing, quality control and partitioning***
Half-hourly means (48 measurements per day), variances, and covariances of greenhouse
gas fluxes, adhering to the principles of Reynolds decomposition, were calculated using
the EddyPro® 7.0.7 software (Li-Cor), according to international standards and protocols
(Sabbatini et al., 2018). The data processing protocol encompassed the following steps:
(1) axis rotation for tilt correction using the double rotation method (Wilczak et al., 2001),
(2) turbulent fluctuations were calculated using block averaging method, (3) time lag was
compensated by covariance maximization with default, (4) Webb–Pearman–Leuning
(WPL) correction of air density fluctuation (Webb et al., 1980), (5) despiking and raw
data statistical screening (Vickers and Mahrt, 1997) and (6) spectral corrections of high-
and low-pass filtering effects. Regarding the latter, high-frequency loss due to path
averaging, signal attenuation and sensor separation was compensated according to
Moncrieff et al., (1997), whereas low-frequency loss due to finite time averaging length
and detrending was corrected according to Moncrieff et al., (2004). Quality check flags
were calculated for flux data according to the widely adopted methodology combining
two tests: steady state test and the developed turbulent conditions test. Over the study
period, we only selected high-quality fluxes (flag value =0) measured when the open-path
sensors were totally clean according to their respective AGC values (AGC value equal to
56 for Open path LI-7500A $CO_2$ /$H_2O$ analyzer and AGC value equal to or higher than
20 for LI-7700.

To quantify the sampling area of flux measurements, a footprint model was

estimated using the method by Kljun et al., (2004) (**Fig 1**). Data periods when the wind
comes from terrestrial adjacent environment (251° to 59°) were rejected, representing





between 45% and 70% of the available daytime and night-time data respectively during
the dry season (GWL=0), and 30% of the available daytime and night-time data
respectively during the wet seasons (GWL>0). Overall, for the nearly 2 years of
measurements, 18% and 8% of the potential daytime data were of good quality for $CO_2$
and $CH_4$ fluxes respectively. Whereas for night-time the available data were reduced to
10 and 5% respectively. The energy balance closure (ratio of the sum of sensible and
latent turbulent fluxes, $H + LE$, to the net radiation minus soil heat flux, $R_n - G$) was 76%
($R^2 = 0.64$; n = 3117).

***Predicting greenhouse gas fluxes as responses to meteorological drivers***
We examine the relationship of $CO_2$ and $CH_4$ fluxes in response to groundwater level,
serving as a proxy for long-term water storage, with air temperature as the main factor
regulating respiration in the system, and incident solar radiation modulating the
photosynthetic rate. Since these variables were measured every 24-hour, we processed
half-hourly $CO_2$ and $CH_4$ fluxes collected to construct 24-hour integrated values. We
selected dates that contained over 50% of the anticipated data points, particularly those
with more than 25 valid measurements distributed during the day, to calculate integrated
daily flux. Selecting 26 and Y days for $CO_2$ and $CH_4$, respectively, well distributed
throughout the measured period (both dry and wet periods). Before integrating daily flux
values, we filled gaps in the half-hourly $CO_2$ and $CH_4$ flux data using linear interpolation
for missing values. This selection criterion aimed to accurately represent the daily pattern
in flux measurement distribution.

To analyze the relationship between integrated daily fluxes and their potential

environmental predictors, we employed a linear regression combined with a forward
model selection technique (Aho et al 2014). This method involved sequentially fitting a
series of regression models, each incorporating different combinations of predictors and
their interactions. The process began with simple models, each containing only one
primary predictor, and gradually increased in complexity to include all possible
interactions among predictors. We then used the Akaike Information Criterion (AIC) to
evaluate and identify the most effective model from the set. The model with the lowest
AIC was selected as the best fit, indicating it provides the most useful balance between
model complexity and explanatory power. After selecting the model, we examined the
influence of the predictors by analyzing the slope $\beta$ coefficients at a significance level of





alpha = 0.05, using a 95% confidence interval to determine if these coefficients were
significantly different from zero.
Additionally, while ground water level (GWL) was measured daily, our model
selection process also aimed to identify which daily measurements of air temperature and
incident solar radiation (mean, minimum, or maximum) were the best predictors. This
method ensured that the chosen model was robust and relevant to the ecological scales
being studied. Detailed instructions on how to run these analyses are detailed in the R
script available in DRYAD (see details in the *data accessibility statement* section).

## 3.  Results

*Time series of greenhouse gas emissions and meteorological drivers*

At Fuente de Piedra Lake, we observed significant seasonal variations in meteorological
conditions, as illustrated in **Fig 2A**. Air temperature ($T_a$) and incident solar radiation
exhibited consistent trends throughout the study period, with mean daily values of $17 \pm 7$
°C for $T_a$ and $18 \pm 8$ MJ m$^{-2}$ d$^{-1}$ for incident solar radiation. In contrast, these two
environmental variables generally followed asynchronous patterns with groundwater
level (GWL) and precipitation events (**Fig 2A** and **B**). Particularly during the summer
months (July, August, September), the highest daily air temperature values coincided with
GWLs beneath the surface and a lack of precipitation. Also, during the "dry" periods, a
salt crust several centimeters thick developed on the sediment (see Figure 3B). For
instance, the minimum $T_a$ recorded was 2 °C in January 2023 corresponding with a period
of frequent precipitation and GWL above the surface. Conversely, the maximum $T_a$ of 34
°C occurred in August 2021, during a period when the GWL was 24 cm below the surface.
We observed a strong correspondence between water storage in Fuente de Piedra
Lake and its capacity to assimilate atmospheric $CO_2$. The $CO_2$ flux patterns can be
categorized into two distinct lake states: a flooded lake (groundwater level >0 cm,
indicated from purple to blue colour in **Fig 2C and D**) and a dry lake (groundwater level
<0 cm, shown from orange to yellow colour). During periods of flooding, the lake acted
as a $CO_2$ sink, with fluxes ranging from 0 to -30 µmol m$^{-2}$ s$^{-1}$. The $CO_2$ assimilation
capacity increases with groundwater level and incident solar radiation, particularly from
January to June. Notably, the flooded periods in the two years of the study showed marked
differences in groundwater level. In 2022, we recorded the highest $CO_2$ uptake of 30 µmol
m$^{-2}$ s$^{-1}$ in May, when the groundwater level was at its peak, 40 cm above the surface. The





peak $CO_2$ uptake in 2023 was approximately half of what was observed in 2022. Although
this peak occurred in March rather than May, it followed a similar trend to the ground
water level (GWL), which was also about half of the peak level observed in 2022 (~20
cm above the surface). In contrast, under dry conditions, Fuente de Piedra Lake cesses
the $CO_2$ uptake, occasionally transitioning to minor $CO_2$ emissions. Notice that during
extreme rainfall pulses within dry periods, when Fuente de Piedra Lake remained
relatively dry with the groundwater level below the surface, we observed notable net $CO_2$
emissions. For example, a heavy rainfall event in September 2021 (36 mm day$^{-1}$) resulted
in $CO_2$ emissions reaching up to 10 µmol m$^{-2}$ s$^{-1}$, with a high elevation of the groundwater
level above the surface (from -20 to near 20 cm). Also, in October of both 2021 and 2022,
subsequent rainfall events (12 mm day$^{-1}$ and 22 mm day$^{-1}$, respectively) corresponded
with $CO_2$ emissions of 7 µmol m$^{-2}$ s$^{-1}$ and 5.5 µmol m$^{-2}$ s$^{-1}$, respectively. In these cases,
emissions occurred under negative GWL conditions.

In the case of $CH_4$, flux was relatively stable throughout the whole study period,
generally acting as a sink, with values fluctuating between -0.2 and 0.1 µmol m$^{-2}$ s$^{-1}$ (**Fig**
**2D**). Furthermore, no clear relationship was observed between $CH_4$ fluxes, and the
meteorological variables examined.

*Fluxes of $CO_2$ and $CH_4$ at a daily scale*
When examining the daily scale during wet periods, it becomes evident that $CO_2$
assimilation predominantly takes place during daylight hours, specifically between 9 am
and 2 pm (local time) (**Fig. 3**). It should be noted that the few emission values observed
within this time frame correspond to the emission occurrences described earlier for the
dry season, which promptly followed rainfall events. In the case of the dry period, it is
noteworthy that a salt crust forms over the lake, leading to a near cessation of $CO_2$ fluxes.
In the case of $CH_4$, there is also a discernible increase in assimilation fluxes between 9
am and 2 pm. This observed pattern is consistent in both wet and dry conditions.

*Model predictions of 24-hour integrated flux values in the study system*
A clear pattern in $CH_4$ flux was not detected during the study period. Likewise, no clear
relationship with the environmental predictors studied could be identified. Additionally,
measurements obtained from the EC tower resulted in a substantial number of gaps in the
$CH_4$ time series, making it impossible to establish a predictive model for these fluxes. On





the contrary, we were able to adjust a robust regression model for $CO_2$ integrated daily
flux. A total of 26 daily integrated $CO_2$ flux values were obtained for the sampling period,
analyzing those dates when more than 25 valid measurements were available. After AIC
model selection routine, the candidate predictive model for daily integrated $CO_2$ flux was
determined to include groundwater level, maximum daily air temperature, and mean
incident solar radiation. Additionally, the model incorporated interactions between
groundwater level and maximum daily air temperature, as well as between groundwater
level and incident solar radiation (Summary Table of the model is available in the
Supplementary Material). Indeed, fitted statistical capacity of the model has a relatively
high explanatory capability (Adjusted $R^2$ =0.72).

The model confirmed a positive effect of groundwater level on enhancing $CO_2$

assimilation in the system (**Fig 4A**; $\beta_{GWL}$ = -0.115, 95% CI = -0.214 to -0.016; note that
assimilation corresponds with negative values of the $CO_2$ flux). While the isolated impact
of air temperature increase on $CO_2$ assimilation could not be determined ($\beta_{Ta}$ = +0.07,
95%CI = -0.053 to +0.192), the model identified an antagonistic interaction between air
temperature and groundwater level ($\beta_{GWL \times Ta}$ = +0.010, 95%CI = +0.005 to +0.015). As
air temperature increases, the positive effect of GWL on $CO_2$ asimilation diminishes (**Fig**
**4B**). Conversely, mean daily incident solar radiation was found to promote $CO_2$
assimilation ($\beta_{Rad}$ = -0.12, 95%CI = -0.21 to -0.03), with a pronounced synergy between
mean daily incident solar radiation and the presence of water ($\beta_{GWL \times Rad}$ = 95%CI = -
0.0116 to -0.003), notably enhancing the capacity for $CO_2$ assimilation (**Fig 4C**).

Using time series data for groundwater level, daily maximum air temperature, and

mean daily incident solar radiation at Fuente de Piedra Lake, we generated retrospective
predictions of $CO_2$ assimilation capacity in the system dating back to 2001 (**Fig 4D**). The
model predictions closely aligned with the observed values for the study period when
using the time series data for the predictors, supporting the robust predictive ability of our
model (Suplementary Material; **Fig S1**). Our estimates indicate a pronounced fluctuation
in $CO_2$ assimilation capacity according to hydrological variations. In years with higher
groundwater level and prolonged water storage the model predicted an exceptionally high
capacity for atmospheric $CO_2$ assimilation of the lake, with annual values surpassing 0.7
Kg C $m^2$ $year^{-1}$ (e.g., in 2011, 2012, 2014, 2020). In contrast, during years marked by
extended droughts, a substantial reduction in $CO_2$ assimilation capacity was modeled.
These drought periods, characterized by dry conditions, resulted in a reduction of the



assimilation capacity to less than a third of the levels recorded in wet years (e.g., from
2006 to 2010).
**4. Discussion**
In agreement with previous research in permanent saline lakes of the Tibetan plateau (Li
et al. 2022), we show that a model Mediterranean shallow saline lake acts as a significant
carbon sink through the uptake of atmospheric $CO_2$ when flooded. Conversely, $CO_2$
assimilation ceases during dry periods. Longitudinal time series analysis reveals that
prolonged droughts indeed hinder the ability of the system to assimilate atmospheric $CO_2$
due to the lack of water, but we also observed that an increase in air temperature during
wet periods moderates the $CO_2$ net assimilation capacity, a process likely related to the
reduction of gas water solubility with temperature. This underscores the pronounced
impact of seasonal and interannual variability, ultimately dictated by drought and rainfall
patterns, on the ability of the studied system to sequester atmospheric carbon. Moreover,
this pattern also displayed considerable variability at the daily scale, closely correlating
with fluctuations in incident solar radiation over daily cycles. In this regard, the $CO_2$
assimilation capacity of the system peaked during those hours of maximum incident solar
radiation. While measurement of $CO_2$ (and $CH_4$) fluxes at multiple scales is challenging
and requires specialized equipment (i.e. eddy covariance sensors), our research proposes
an alternative proxy. By integrating data from environmental predictors at various scales,
we have achieved highly accurate predictions of $CO_2$ exchanges between Fuente de
Piedra Lake and the atmosphere. In essence, we estimate $CO_2$ flux through the continuous
measuring of accessible environmental variables, namely, the amount of water, air
temperature, and incident solar radiation.
In shallow, well-mixed, and oxygenated systems like Fuente de Piedra Lake, the
photosynthetic capacity of the phytoplankton community is closely linked to the water
column height (i.e. groundwater level) (Batanero et al. 2017), promoting $CO_2$ assimilation
as the extent of the habitat for these communities expands (Wetzel, 2001). Related to the
aforementioned, a significant synergy exists between water storagein the ecosystem and
incident radiation, serving as a proxy for the photosynthetically active radiation upon
which photosynthesis depends. This interaction occurs on both a daily scale, associated
with variations in light intensity following day-night cycles, and an annual seasonal scale,
largely determined by changes in daylight hours throughout the year. Notably, during the



night, the net exchange of $CO_2$ between the water and the atmosphere in Fuente de Piedra
Lake is negligible. This could be attributed to the absence of photosynthesis during
nighttime. Additionally, the high salinity inherent to these environments constrains
methanogenesis, which is the least energy-efficient carbon mineralization process in the
redox sequence (reviewed in Soued et al., 2024). Considering the above, it appears to
offer a plausible explanation for why microbial respiration does not surpass inorganic
carbon assimilation through photosynthesis in systems like Fuente de Piedra Lake during
wet periods, despite the high content of dissolved organic carbon (Batanero et al., 2017).
What is more, despite the lack of $CH_4$ flux data, our results position Fuente de
Piedra lake as a $CH_4$ sink. Vast approximation estimates determined that Fuente de Piedra
could uptake on average 1.83 mg C m$^{-2}$ day$^{-1}$ and 3.70 mg C m$^{-2}$ day$^{-1}$ during the wet and
the dry period respectively (Suplementary Material; **Fig S2**). Such values are even higher
than those measured in typical well aerated soils such us soil forest or grasslands, with
average rates of 0.4-1.26 mg C m$^{-2}$ day$^{-1}$ (Murguia-Flores et al., 2021; Perez-Quezada et
al., 2021). The double value of uptake during the dry periods compared to the wet ones
appear to be consistent with some proposed mechanisms promoting $CH_4$
reductionaccording to the existing literature, since the increase of temperature together
with gas diffusivity due to loss of water, may increase methane oxydation in a similar
way to terrestrial ecosystems (Chen et al., 2010; Rafalska et al 2023). However, caution
is needed when interpreting our results, as the dynamics of methane fluxes could become
very complex in an intermittent system like Fuente de Piedra. On the one hand, just as
methanogenic activity is inhibited by salinity (Herbert et al., 2015), methanotrophic
activity has also been observed to be significantly reduced by salinity in terrestrial
systems (Ho et al., 2018). However, methane oxydation processes associated with aquatic
prokaryotes may be more resistant to salinity (Khmelenina et al., 2010; Deng et al. 2017),
especially if the variation is gradual (Osudar et al., 2017). Thus, further measurements
and analysis are needed to estimate the role of methane oxydation and the relevance of
saline intermitent lakes as $CH_4$ sinks in a climate change scenario.
Drought periods are accompanied by an increase in air temperature, with the high
air temperatures recorded immediately before the system completely dries out. We have
found that this rise in air temperature leads to a reduction of the system's capacity to
assimilate $CO_2$, even during wet conditions. A direct consequence of climatic warming is
the reduction of gas solubility accentuated in saline wetlands (Batanero et al. 2022) . In
addition, increase in temperature can enhance microbial metabolic rates and therefore,





biomass-specific $CO_2$ production (Smith et al. 2019). Given that endorheic saline lakes
are fueled by significant amounts of organic matter (Li et al., 2017; Batanero et al., 2017;
Song et al., 2013), it is unsurprising that warming leads to a decrease in net primary
production in the system as a result of enhanced microbial respiration, and consequently,
a reduction in $CO_2$ assimilation capacity. In addition, carbon emissions in inland waters
could increase with warming, independently of organic carbon inputs, simply because the
apparent activation energy is predicted to be higher for respiration than photosynthesis
(Yvon-Durocher et al. 2010; Yvon-Durocher et al. 2012). Finally, it has been recognized
that photosynthesis is often the first process to be affected by environmental stressors,
with photosynthetic capacity diminishing prior to other cellular functions (Feller, 2016;
Cardona et al., 2018). Specifically, carbon assimilation through the Calvin–Benson cycle
exhibits particular vulnerability to both drought and elevated temperatures, occurring
even when photosynthetic electron transport continues to operate effectively (Sharkey,
2005). On the whole, we show the profound synergy between global warming and
intensifying drought severity and frequency, disrupting the $CO_2$ assimilation capacity of
Mediterranean saline lakes and leading to negative feedback loops.

## 418   5. Conclusion

While the desiccation of saline lakes is not novel, with researchers highlighting
the concerning increase in dry periods within many of these ecosystems over recent
decades (Williams 1993; Gross 2017; Wurtsbaugh et al. 2017; Wang et al. 2018), our
study underscores the significant implications this has for the ecosystem services they
support. Our retrospective predictions show that in wet years, the system could exhibit a
high $CO_2$ assimilation rate. For instance, between 2010 and 2015, we estimated that
Fuente de Piedra Lake had an average assimilation rate of 0.83 (SD = ±0.27) kg C m$^{-2}$
year $^{-1}$, comparable to the net assimilation observed in evergreen or deciduous forest
systems (Pastorello et al., 2020). This result challenges the generalised belief that inland
waters primarily act as sources of greenhouse gases (Raymond et al. 2013). Conversely,
the system undergoes significant reductions in its annual atmospheric $CO_2$ sequestration
capacity during dry periods. For instance, under severe drought conditions as observed in
Fuente de Piedra from 2005 to 2009, the annual $CO_2$ sequestration is estimated to have
fallen to less than a quarter of what was observed in more humid periods. Climate change
projections, including even the most optimistic scenarios, forecast an increase in both the



frequency and duration of heatwaves and droughts in the coming years (Trenberth 2011,
Perkins-Kirkpatrick 2020). This implies that saline lake ecosystems in arid and semi-arid
endorheic basins will remain dry for longer periods, or may even vanish, resulting in the
loss of a significant carbon sequestration pathway. Importantly, the disappearance of
saline lakes due to water scarcity has been largely attributed to anthropogenic water
overuse (i.e., agriculture) rather than to macroclimatic phenomena (Wurtsbaugh et al.
2017). This seems to be the case of Fuente de Piedra Lake, as the catchment area is
dominated by agricultural land. Thus, a proper water management during drought periods
seems to be the most plausible solution to preserve the ecosystem services provided by
Mediterranean saline lakes.


**Author contribution**
PS-O and IR conceived the study; all authors contributed to the installation and
maintenance of the eddy covariance tower; IA led the fieldwork and the processing of the
samples with the help of the rest of the authors; IP-M carried out the analyses and the
preparation of the results; IA, IP-M, and PS-O conducted the preparation of the first draft
of the work; all authors participated in the drafting of the final draft.

**Data accessibility statement**
The R script used to conduct the data analysis and the datasets are available at the Dryad
Digital Repository:
https://datadryad.org/stash/share/qEpPRJopVR132UszL3bnaxoZh07ADL0E5LpVL6xC
SZA


**Financial support**
This work was partially support by  the projects PID2020-117825GB-C21 and PID2020-
117825GB-C22 funded by MCIN/AEI/10.13039/501100011033, LifeWatch-2019-10-
UGR-01 and LifeWatch-2019-09-CSIC-13 funded by the MCIN through the FEDER
funds from the Spanish Pluriregional Operational Program 2014-2020 (POPE),
LifeWatch-ERIC action line and project BAGAMET (P20_00016) funded by the
Counseling of Economy, Innovation, Science and Employment from the Government of
Andalucía, including European Union ERDF funds. I.P.-M. developed his research as





part of the eWARM project, supported by the Marie Skłodowska-Curie postdoctoral
fellowship 2022 (project number 101110111).

**Competing interests**
The authors declare that they have no conflict of interest.

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



FIGURES

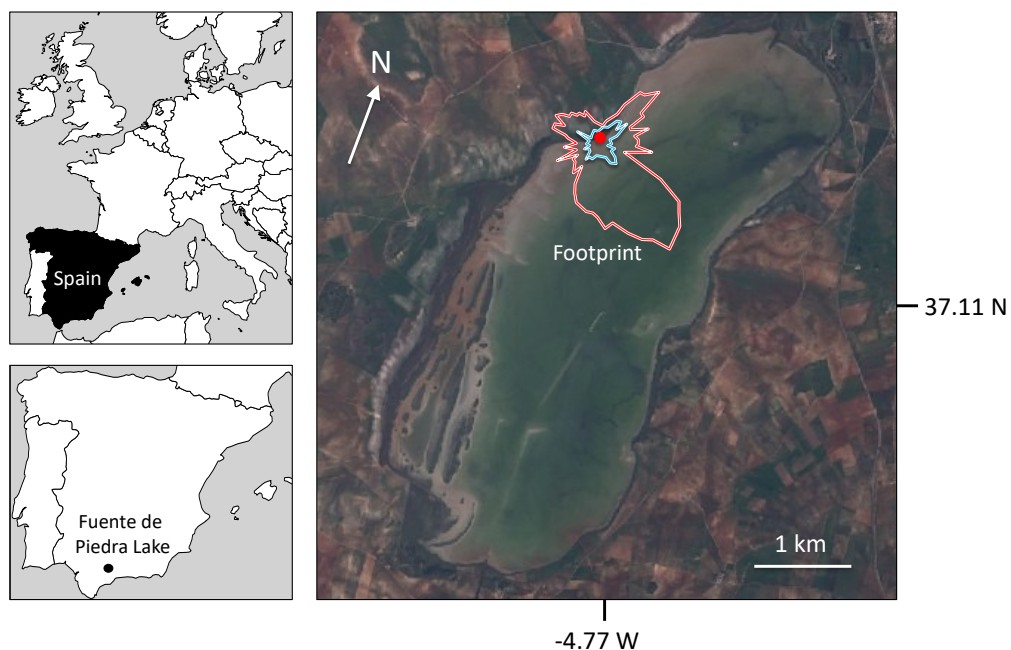

**Fig 1.** Location of the Fuente de Piedra Lake (province of Málaga, South of Spain). Dot inside polygon in right panel shows the location of the eddy covariance tower. The areas within the footprint contributing the 90% to measured fluxes are delimited inside polygons for daytime (blue) and nighttime (red).





**Fig 2.** Time series of (A) air temperature and incident solar radiation, (B) groundwater level and precipitation (PPT), (C) $CO_2$ and (D) $CH_4$ flux, collected at Fuente de Piedra Lake during 2021, 2022, and 2023.





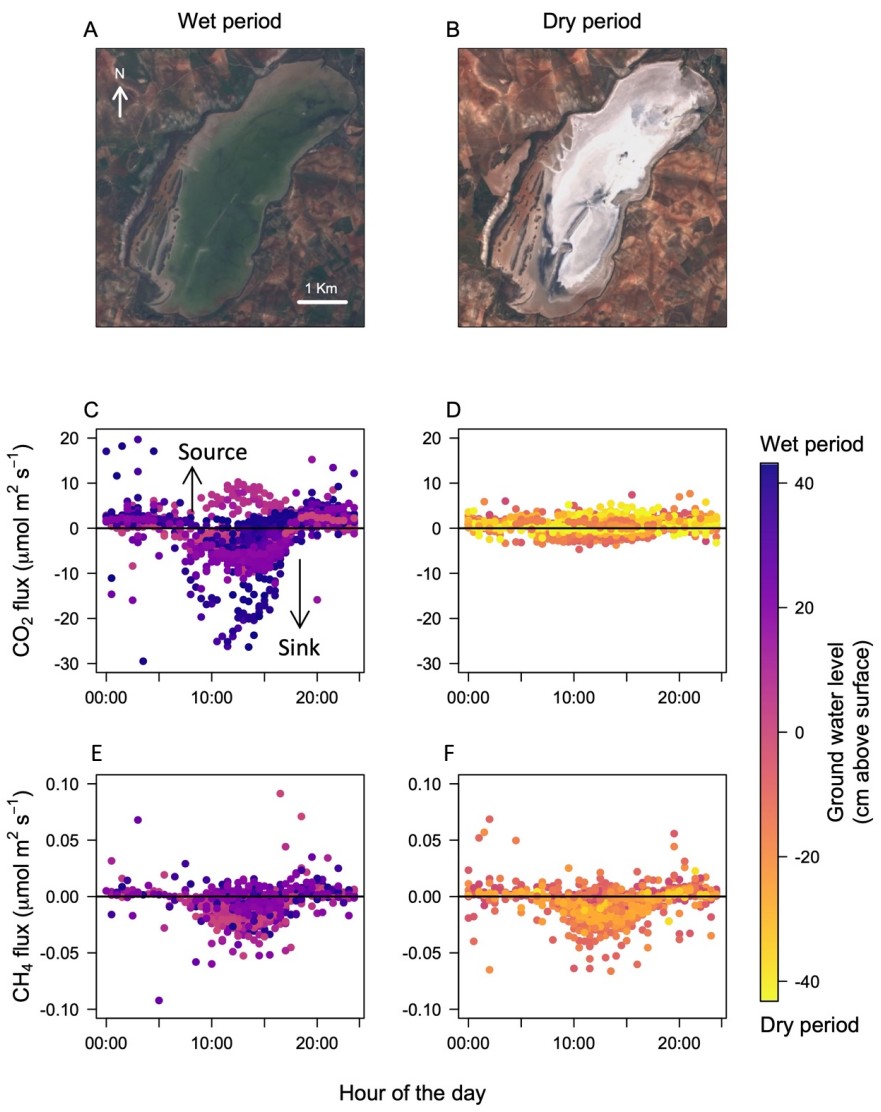

**Fig 3.** Aerial photos of Fuente de Piedra Lake during a period of maximum water availability (wet period, **A**) and during a typical dry episode (**B**). Note that for some period during the dry episodes, a salt crust forms covering practically the entire extent of the lake. The figure shows the daily pattern of $CO_2$ and $CH_4$ fluxes during the wet period (**C** and **E** respectively) and the dry period (**D** and **F** respectively). Water availability is measured in terms of groundwater level.





**Fig 4.** Prediction of $CO_2$ flux as a response to groundwater level (**A**), the interaction between groundwater level and daily maximum temperature (**B**), and the interaction between groundwater level and incident solar radiation (**C**). Using existing time series for the model predictors, it has been possible to reconstruct the estimated $CO_2$ fluxes, as well as the annual cumulative value of $CO_2$ removal since 2001 for Fuente de Piedra Lake (**D**).