# Peer review of "Drought conditions disrupt atmospheric carbon uptake"

_EGUsphere, 2024_

## Author Response (AR1)

Please, see below the response to the reviewer and the actions taken regarding their concerns. We also included a PDF version of the revised manuscript with changes highlighted in yellow.

**REVIEWER 1**

**GENERAL COMMENT**
**They find the importance of keeping the lake wet in terms of inducing CO2 uptake, and then discuss some implications for lake management in the face of climate change. Some better modeling approaches (e.g. more rigorous cal/val, better statistical descriptions of regression model fits) and a clearer sense of the uncertainty of these models derived from sparse data would boost confidence in the paper's findings.**

R: We appreciate the thorough and constructive review from Reviewer 1. The reviewer has identified critical weaknesses in our work that required attention. Notably, many of these issues also align with those highlighted by Reviewer 2. After addressing all the concerns raised, we are confident that the manuscript has improved significantly in robustness, clarity, and overall quality.

**MAJOR COMMENTS (MCS)**
**The paucity of data collected is challenging and under-described. More attention should be given to this challenge. (e.g. how it came to be, and how the results can still stand even with such a small dataset).**

R: This issue has also been highlighted by Reviewer 2. Much of the problem stems from a lack of clarity in explaining our data in the text.

Firstly, our emissions database is indeed extensive, with 4128 measurements for $CO_2$ and 2425 for CH4, providing a robust capture of system flux patterns during the study period (see Fig. 2 and Fig. 3). Gaps in the time series only affect the number of days available for estimating daily $CO_2$ emissions.

It is important to note that gaps in long-term data records are inevitable when using eddy covariance, primarily due to instrument failure and insufficient turbulence (Baldocchi et al., 2003; Falge et al., 2001; Aubinet et al., 2012). Data coverage is further reduced when wind originates from undesirable sectors (land inour case), though this does not compromise measurement quality (Goloub et al., 2023).

Despite these limitations, most studies derive their primary results from annual or seasonal $CO_2$ and CH4 balances using various gap-filling procedures. In contrast, we are very restrictive in quantifying daily fluxes due to limited data coverage. Our main results are derived from a predictive model developed using data from measured days with relatively good coverage:

*"We selected dates that contained over 50% of the anticipated data points, particularly those with more than 25 valid measurements well-distributed throughout the day, to calculate integrated daily flux" (L231-L233).*

Lastly, our regression model, despite being fitted with a limited dataset, allows for accurate predictions due to the well-distributed data throughout the time series. The model's high $R^2$ value ($R^2$=0.73), achieved with few predictors, exhibits its robustness and effectiveness. Additionally, it is worth noting that our model can detect the high variability in fluxes observed during very wet periods as a result of interaction with environmental variables (compare the dispersion in the raw data for wet months in Fig 2 with the predictions for wet periods Fig 4).

**ACTIONS TAKEN:** We agree with the reviewers that the nature of our data needs to be better explained. Therefore:

L223-233: We have revised the methodology to explain in detail the limitations of the eddy covariance technique concerning the gaps generated in the time series. Additionally, we included a brief justification for our approach in calculating daily fluxes.

L278-280: We included information on the exact number of measurements obtained with the eddy covariance and their coverage during the study period.

L327-329: We state the number of valid measurements used in our model of daily $CO_2$ flux after applying our restrictive criteria.

**The results presented in L322-335 are for modeling retrospectively the $CO_2$ assimilation capacity since 2001; none of this is given as an objective nor is it in the methods.**

R: Agree. Modeling retrospectively the $CO_2$ assimilation capacity requires much greater emphasis in the text than we have given it in the original version.

ACTIONS TAKEN: As the reviewer suggested, we have included the objective of retrospectively modeling $CO_2$ at the end of the introduction, along with the justification and rationale for this (L109-111; 114-115). Additionally, we have provided a detailed explanation of the retrospective $CO_2$ modeling in the methodology section (L256-L259).

**The site's hydrology and its implication for spatial and not just temporal patterns of drought is not well described. When the GWL sensor reads negative, what fraction of the lake is not inundated, and vice versa?**

R: Point taken. Fuente de Piedra Lake is very shallow and flat throughout, with the groundwater level (GWL) remaining relatively constant across the system. Nevertheless, the piezometer is located in the central and deepest part of the lake, so negative GWL measurements indicate that the entire system lacks surface water.

ACTIONS TAKEN: We have included a brief explanation of the piezometer's location in the text (L172-L176). Additionally, we have added its location to Fig. 1.

**MINOR COMMENTS**
**The title is a bit dramatic; perhaps "Drought conditions disrupt…" (particularly since the drought conditions may derive from irrigation or water resources management as much as meteorology).**
> R: Point taken. We edited the title accordingly.

**In the supplemental document, Fig 1 and 2 can have statistics added (r2, etc.)**
> R: Done.

**L68 consider describing here the mechanism of uptake – is it abiotic dissolution from partial pressure differences or plant/algae/diatom uptake as part of their primary productivity?**
> R: Added (L68)

**L155 consider adding whether these data were logged on a 7550 AIU or a Campbell logger, or wherever.**
> R: Data were logged on a Campbell logger (L158-L160)

**L164 is the soil and lake heat storage considered? It should be. (I'd also tend to say heat flux instead of heat flow)**
> R: Thanks to the referee for this comment that allows us to clarify the objective of the energy balance closure in this work and clarify how we have done it. Since this study is not about the energy balance of FdP, the purpose to show the results of the energy balance closure is to provide additional information regarding the turbulent flux quality and demonstrate that the eddy covariance instruments are working properly. For this purpose, the energy balance closure was done during the drought period (lake heat storage= 0 and soil water content negligible) and without excluding data due to wind direction. ACTIONS TAKEN: We have included a brief explanation clarifying the objective of the energy balance closure and how we proceded (L211-216)

**L169 clarify that the GWL is also the lake height sensor(?)**
> R: Added. Now (L176).

**L204 comment on why there is so sparse a dataset! 8 and 18% of flux values over this period is very poor data coverage.**
> R: Please, see our response to mayor comment 1.

**L206-8, consider daily closure, and including storage terms**
> R: We have added that the soil heat storage term is included in de definition of G (L166-L167). Since the objective of the energy balance closure was to provide additional information regarding the turbulent flux quality, we have calculated the

energy balance closure using the half hour data, following Wilson et al 2002, in order to compare with the turbulent flux quality of others FLUXNET sites.

**L297 at what scale is no pattern detected (because the previous sentence says "this…pattern is consistent…").**
R: This sentence was poorly formulated and has been revised. It now reads: 'No evident relationship between $CH_4$ flux and the environmental predictors studied was found during the study period'. (Now in L308-310).

**L303 it's not clear how the half-hourly values were integrated? Just connect the dots, or some advanced gap-filling?**
R: Trapezoidal integration of the values measured every 30 minutes was performed to calculate the daily flux. We added this explanation in L235-237.

**L340 notes the C sink but how about the release?**
R: Point taken. In the new version, we have also discussed the cases of emission, which are mostly observed immediately after rainfall episodes following the dry period (See Fig 1). (L366-368).

**L354 "highly accurate predictions" seems like an exaggeration given the lack of cal/val and rigor in the modeling**
R: We toned down our statement following reviewer comment (L382-384)

**L374 how high is the DOC here**
R: Range of DOC added (L404).

**L375 be more nuanced or less total rather than "the lack of data"**
R: Point taken. We edited accordingly (L405-406).

**L375 "vast approximation estimates" is unclear**
R: Point taken. We clarified the use of rough estimates (L406).

**L387 some more detail about the intermittency of this system vs others would be helpful.**
R: Done. Details have been added in the Material and Methods section, when introducing the system (L139-141).

**Fig 1 lacks a source for the image; I'd also wonder if a photo of the tower and the surroundings could help the reader understand the landscape and measurement system better.**
R: Source added in the figure caption (Sentinel-2). We also added a photography in the supplementary material showing the installed tower next to the first author.

**Fig 3 A,B lack a source for the images (and when are they?); consider adding "n" to C-F – how many data points are we seeing?**
R: Point taken. We specified sampled size in the figure caption.

**Fig 4 A should have r2, slope, etc.**

R: We believe that our figures are already busy, so we have added the information requested by the reviewer in the figure caption. Please, note this information is in the text too.

**Fig 4 Figs B and C are difficult to read and it may be easier to see a family of curves on a 2D graph than a surface on a 3D graph.**

R: We prefer to represent the surface in 3D because the interaction occurs between continuous variables. However, we understand that 3D graphs can be confusing. Therefore, we have included a small panel in the corner of the 3D plots explaining the process represented continuously, but only taking the extreme values.

**TECHNICAL COMMENTS**

**L52 change they account to accounting**

R: Done

**L64 change The first to Past**

R: Done

**L69 change need it to needed**

R: Done

**L105 perhaps "in Spain's saline lake Fuente de Piedra"**

R: Done

**L129 check spelling of plankton**

R: Done

**L147 add "the" after Thus,**

R: Done

**L195 remove totally; define AGC; I suspect that AGC can be equal to or above 56, but don't know…**

R: Done. AGC = Automatic Gain Control.

**L200 add the before terrestrial**

R: Done

**L205 reword (whereas is used poorly)**

R: Done

**L218 perhaps Y is an error; there is also no verb in this sentence.**

R: This sentence was a typo. It has been removed.

**L279 ceases not cesses**

R: Done

**L362 check space after storage**

R: Done

**L379 as not us**

R: Done

**L381 reword "the double value"**

R: Done

**L383 space after reduction**

R: Done

**L399 extra space after citation**

R: Done

**L400 add "an" before increase**

R: Done

**L422 add approach before has; replace they with these landscapes**

R: Done

**L441 add system after management**

R: Done
* * *
**REVIEWER 2**

**GENERAL COMMENT: Overall comment: The paper is well written and easy to follow. The introduction provides a good background/motivation for the study, explains current research gaps and clearly states the objectives for their study. I think the authors should more thoroughly discuss the impact of rejecting up to 95% of the data. Also, I think the figures could benefit from some revisions (see comments below), and there is some confusion about the supplementary figures (for example, the numbers don't seem to match the descriptions in the main text). Lastly, it would be nice to see a discussion of your results in a broader context.**

R: We appreciate the supportive words regarding the quality of our work and the constructive revision provided. Like Reviewer 1, Reviewer 2 emphasizes the need to present our data more clearly. We also agree with the suggestion to further explore the broader implications of our work and improve the figures. Therefore, we have incorporated all the comments proposed by the reviewer. Detailed explanations of the changes based on the reviewer's comments are provided below.

**Abstract: It could benefit from the inclusion of specific quantitative results – e.g., by how much is the carbon uptake increased/decreased? What is the magnitude of the "significant carbon sink"?**

> R: Agreed. We have added specific quantitative results in the abstract as proposed (L30-36).

**Line 69: Remove the word "it"**

> R: Done.

**Lines 203-206: The authors state that only 8%-18% ($CO_2$) and 5%-10% ($CH_4$) of the data "were of good quality", which means the majority of data collected over the 2-year sampling period was rejected. Are these values common for these types of studies? What are the implications of rejecting the majority of the data?**

> R: This comment matches with the main major comment proposed by Reviewer 1. Please refer to our response to MC-1 for more details.
>
> Briefly, our emissions database is indeed extensive, with 4128 $CO_2$ and 2425 $CH4$ measurements well distributed on time, providing robust system flux patterns during the study period (see Fig. 2 and Fig. 3). Gaps in the time series only affect the number of days available for estimating daily $CO_2$ emissions.
>
> Note that, gaps in long-term data records are inevitable with eddy covariance due to instrument failure and insufficient turbulence (Baldocchi et al., 2003; Falge et al., 2001; Aubinet et al., 2012). Data coverage is further reduced when wind originates from undesirable sectors, but this does not compromise measurement quality (Goloub et al., 2023). Therefore, most studies use gap-filling procedures for annual or seasonal $CO_2$ and $CH_4$ balances. In contrast, we restrict our daily flux quantification to days with over 50% data coverage and more than 25 valid measurements distributed throughout the day (L216-L218). Additionally, our model detects the high variability in flux during very wet periods due to interaction with environmental variables (compare the dispersion in raw data for wet months in Fig 2 with the predictions for wet periods).
>
> **ACTIONS TAKEN:** We agree with the reviewers that the nature of our data needs to be better explained. Therefore:
>
> L223-233: We have revised the methodology to explain in detail the limitations of the eddy covariance technique concerning the gaps generated in the time series. Additionally, we included a brief justification for our approach in calculating daily fluxes.
>
> L278-280: We included information on the exact number of measurements obtained with the eddy covariance and their coverage during the study period.
>
> L327-329: We state the number of valid measurements used in our model of daily $CO_2$ flux after applying our restrictive criteria.

**Line 218: "Selecting 26 and Y days for $CO_2$ and $CH_4$, respectively" – I don't understand what this means.**

> R: this sentence was a typo. We deleted this sentence from the text.

**Lines 250-252: Can you briefly mention why there is a peak in PPT and GWL in September of 2021? This contrasts your statement about high temperature coinciding with lack of precipitations during summer months.**

> R: We mentioned the brief peak in precipitation and an increase in GWL between August and September 2021 as an exception in the general pattern (L267-270).

**Lines 254: The text says the minimum temperature recorded was in January 2023, but according to Figure 2, this looks like December 2022.**

> R: The minimum temperatures are very similar in December 2022 and January 2023. However, it is correct that they reach lower values in January 2023, as seen in Figure 2 (orange line in panel A).

**Line 265: May 2022 records the highest $CO_2$ uptake, but also the highest $CO_2$ outgassing. Both the sink and source show fluxes around 20 μmol. Wouldn't these values almost cancel each other out, resulting in a much smaller net sink? Could you also report absolute flux values for certain periods, and not just the max uptake and max outgassing? On Figure 2 panel C and D, could you add a line representing the mean absolute flux, above the spread of flux measurements?**

> R: We believe that part of the problem arose because the Y-axis of panel C was not symmetrical (max at 20, min at -30). Despite the significant variability observed in May 2022, the net $CO_2$ flux value is indeed negative (-3.11 μmol m$^{-2}$ s$^{-1}$). Thus, we have edited the figure to ensure the axes are symmetrical, making it much more evident that the net uptake value for the wet period exceeds the outgassing values. Also, as the reviewer suggested, we have provided absolute values for the periods of highest uptake in this new version including some more information about the great variability observed in $CO_2$ flux (L287-295 for $CO_2$, L306-309).
>
> Please, note that the annual cumulative value (annual net uptake) is also provided in the last panel of Figure 4. Additionally, the reviewer can verify that our predictions can capture this significant variability during the wet periods due to the interaction between predictors (see Fig 4).
>
> Lastly, regarding the reviewer's recommendation to include the annual average value as a line, we have chosen not to do so. Our figures are already quite dense, and the intention of Figure 2 is to present our raw data. Drawing the trend of the average over time would require interpolating missing values.

**Line 297: The fact that there is no clear pattern in $CH_4$, does this have to do with the fact that up to 95% of the data was removed?**

> R: Not in this case. The data loss primarily affects how we calculate the daily flux, which is used for predictive models. At this point, we are attempting to relate the pattern of the raw data with the environmental variables we have measured, but no clear patterns are evident at first glance. However, to avoid misinterpretation by readers, we have edited this section to make it clear that

the observed data show a pattern that does not seem to align clearly with the predictors (L308-310).

**Line 308: I can't find the "Summary Table" in the Supplementary Material. Does this table contain more error statistics?**

R: Thus is a Typo, summary table is not provided as we report coefficients and 95%CI in the text. Also, in this new version $R^2$ of the model is also included in the text (see L336-346).

**Line 310: Is this R value referring to the results of Supplementary Figure S2?**

R: $R^2$ here refers to the fitted model (Fig 4 top panels). We specified this in the new version (L334). Also, note that $R^2$ has been added to all supplementary figures.

**Line 327: You refer to Supplementary Figure S1 here, but should it be S2?**

R: This was a typo. We solved this issue.

**Lines 338-341: Could you add quantitative results here? In parentheses after "significant carbon sink" add actual values of the carbon uptake. What are the flux values for the other saline lakes you refer to? By how much does the $CO_2$ sink decrease?**

R: Added (L365-369).

**Line 378: You refer to Supplementary Figure S2 here, but it should be S3?**

R: solved (L408).

**Overall comment about the Discussion:**
**It would be nice to include a paragraph discussing what your sink values actually mean in a greater context – compare with other sinks. For example, how large is this sink specifically compared to for example other lakes, and/or other land/ocean sinks? Is it possible to use your results to make some upscale estimations representing a larger region, or do you assume that your results are very specific for this lake?**

R: We believe this comment is very insightful and constructive. In the new version of the manuscript, we have included a new paragraph in the discussion that frames our results within a greater context (L447-463).

**Figure 2:**
**Instead of the vertical dashed lines at two of the Januarys, could you differentiate dry and wet periods on all the panels? Or winter vs. summer. For example, have light gray (wet period/summer) and darker gray (dry period/winter) shadings in the background spanning the appropriate months.**

R: On this point, we disagree with the reviewer and prefer not to include the proposed qualitative distinction. In our work, the dry or wet conditions of the system are defined by the GWL value as a continuous variable. This is important because it assigns a quantitative value to the GWL. For this reason, we use a color gradient from yellow to purple throughout the work based on the GWL value, treating GWL as a covariate in our models. Defining the periods as a qualitative

or seasonal factor would not be entirely accurate, as there are varying degrees of "wet period."

However, we recognize that the word "period" can be problematic as it connotes specific dates. Therefore, we have simplified the terminology to "wet" and "dry" as the extremes within the gradient. Also, we replaced "wet/dry period" by "wet/dry conditions".

**A: For clarity, could you match the Temperature and Radiation y-axis values with the corresponding colors shown in the figure? So, orange for the Temperature and green for the Radiation.**

R: Good point. We edited the color of y-axis accordingly.

**B: I assume 0 corresponds to the surface, right? So, the description of the y-axis is a bit confusing, since anything below 0 should be below surface, right? This also applies to the color bar in C and D.**

R: The reviewer is right. We removed "above surface" from the Y-label and we explained that the horizontal line is the surface within the figure caption. Note that we also modified this in the rest of the figures.

**Also, as suggested for A, could you give the precipitation y-axis values a blue color matching the graph?**

R: Done.

**Figure 3:**
**I know the color bar differentiates between dry and wet periods, but I suggest adding headers above the panels for additional clarity. For example, something like this: above the top left panel: "Wet period" and above the top right panel: "Dry period".**

R: Done.

**For clarity, I would move the arrows, indicating source and sink, outside the plot. For example, place the arrows on the right-hand side of panel C, above and below the horizontal line at 0.**

R: Done.

**Figure 4: I find it really hard to interpret panel B and C.**

R: We added two small panels in the right corner of each 3D plot to assist the reader in interpreting the surface plot. However, please keep in mind that this remains the most accurate way of representing interactions across continuous variables.

**Figure S1: Is this figure legend correct? Is the y-axis description correct?**

R: The caption was wrong. We corrected figure caption in Fig S1.

**Figure S2: Could you add a R value on the figure?**

R: Done.

**It seems like you need to update your text with regards to the numbering of the supplementary figures. You only refer to Figs. S1 and S2 (S3 is not mentioned anywhere). Also, I think when you refer to S1 in the text, you mean S2, and when you refer to S2 you mean S3. That means that there is no reference to Fig. S1 in the text. I also think Fig. S1 has the wrong legend. So, it is a bit unclear to my what Fig. S1 is supposed to show.**

R: Solved.

**REFERENCES**

Aubinet, M., Vesala, T., and Papale, D. Eddy covariance: A practical guide to measurement and data analysis. London, UK Springer, 2012.

Baldocchi, D. D. Assessing the eddy covariance technique for evaluating carbon dioxide exchange rates of ecosystems: past, present and future. Glob. Chan. Biol. 9(4), 479-492, https://doi.org/10.1046/j.1365-2486.2003.00629.x , 2003.

Golub, M., Koupaei-Abyazani, N., Vesala, T., Mammarella, I., Ojala, A., Bohrer, G., et al. Diel, seasonal, and inter-annual variation in carbon dioxide effluxes from lakes and reservoirs. Environ. Res. Lett. 18(3), 034046

Falge, E., Baldocchi, D., Olson, R., Anthoni, P., Aubinet, M., Bernhofer, C., ... & Wofsy, S. Gap filling strategies for long term energy flux data sets. Agric. For. Meteorol. 107(1), 71-77, https://doi.org/10.1016/S0168-1923(00)00235-5 , 2001.

---

## Author Response (AR2)

**UNIVERSIDAD**
**DE GRANADA**

3 October 2024

Dear Editorial Committee,

On behalf of all the co-authors of this work and myself, we would like to sincerely thank you for the thorough review process. We have also made the minor editorial changes that were suggested. Additionally, we deeply appreciate seeing our work virtually accepted in such an excellent and prestigious journal.

Kind regards,
Ignacio Peralta-Maraver